# Understanding perceptions of schistosomiasis and its control among highly endemic lakeshore communities in Mayuge, Uganda

**Lazaaro Mujumbusi** [1,2]*, **Edith Nalwadda**[1], **Agnes Ssali**[1], **Lucy Pickering**[2], **Janet Seeley**[1,3], **Keila Meginnis**[2,4], **Poppy H. L. Lamberton**[4,5]*

**1** Medical Research Council/Uganda Virus Research Institute & London School of Hygiene and Tropical Medicine, Uganda Research Unit, Entebbe, Uganda, **2** Institute of Health and Wellbeing, University of Glasgow, Glasgow, United Kingdom, **3** London School of Hygiene and Tropical Medicine, London, United Kingdom, **4** School of Biodiversity, One Health & Veterinary Medicine, University of Glasgow, Glasgow, United Kingdom, **5** Wellcome Centre for Integrative Parasitology, University of Glasgow, Glasgow, United Kingdom

* Lazaaro.Mujumbusi@mrcuganda.org (LM); Poppy.Lamberton@glasgow.ac.uk (PHLL)

**Data Availability Statement:** Due to the nature of the data and the informed consent obtained, in which participants were assured of anonymity, we

## Abstract

### Background

Schistosomiasis is a neglected tropical disease and a serious global-health problem with over 230 million people requiring treatment, of which the majority live in Africa. In Uganda, over 4 million people are infected. Extensive parasitological data exist on infection prevalence, intensities and the impact of repeated praziquantel mass drug administration (MDA). However, how perceptions of schistosomiasis shape prevention and treatment practices and their implications for control measures are much less well understood.

### Methods

Rapid ethnographic appraisals were performed for six weeks in each of three *Schistosoma mansoni* high endemicity communities on the shores of Lake Victoria, Mayuge District, Uganda. Data were collected between September 2017 and April 2018. Data were collected through structured observations, transect walks, and participant observation, and sixty in-depth interviews and 19 focus group discussions with purposively recruited participants. Data were analyzed thematically using iterative categorization, looking at five key areas: perceptions of 1) the symptoms of schistosomiasis; 2) the treatment of schistosomiasis; 3) how schistosomiasis is contracted; 4) how schistosomiasis is transmitted onwards and responsibilities associated with this; and 5) how people can prevent infection and/or onward transmission.

### Results

Observations revealed open defecation is a common practice in all communities, low latrine coverage compared to the population, and all communities largely depend on lake water

cannot include all of the raw data, as some information may jeopardise the anonymity of participants. Therefore, the data upon which this paper is based have been included in the Supplementary information with all reference to names, household locations, and specific job identifiers removed.

**Funding:** This study was funded by a Medical Research Council (https://mrc.ukri.org/) Global Challenges Research Fund Award (MR/P025447/1) to primary investigator PHLL, and co-investigators LP and JS. PHLL is also funded by a European Research Council (ERC) (https://erc.europa.eu/) Starting Grant (SCHISTO_PERSIST 680088), the Engineering and Physical Sciences Research Council (https://epsrc.ukri.org/) (EP/R01437X/1 and EP/T003618/1) and the Wellcome Trust (https://wellcome.org/) [204820/Z/16/Z]. The funders played no role in the study design, data collection and analysis, decision to publish, or preparation of the manuscript.

**Competing interests:** The authors have declared that no competing interests exist.

and contact it on a daily basis. Perceptions that a swollen stomach was a sign/symptom of 'ekidada' (caused by witchcraft) resulted in some people rejecting free praziquantel in favour of herbal treatment from traditional healers at a fee. Others rejected praziquantel because of its perceived side effects. People who perceived that schistosomiasis is caught from drinking unboiled lake water did not seek to minimize skin contact with infected water sources. Community members had varied perceptions about how one can catch and transmit schistosomiasis and these perceptions affect prevention and treatment practices. Open defecation and urinating in the lake were considered the main route of transmission, all communities attributed blame for transmission to the fishermen which was acknowledged by some fishermen. And, lastly, schistosomiasis was considered hard to prevent due to lack of access to safe water.

## Conclusion

Despite over 15 years of MDA and associated education, common misconceptions surrounding schistosomiasis exist. Perceptions people have about schistosomiasis profoundly shape not only prevention but also treatment practices, greatly reducing intervention uptake. Therefore, we advocate for a contextualized health education programme, alongside MDA, implementation of improved access to safe-water and sanitation and continued research.

### Author summary

Despite the World Health Organization's (WHO's) supported mass drug administration (MDA) campaigns, schistosomiasis remains a major global-health concern, especially in resource-constrained countries with inadequate access to safe-water and sanitation, such as Uganda. However, uptake of control interventions including praziquantel MDA is far below the WHO recommended coverage of at-least 75%. We explored community perceptions of schistosomiasis and how these affected control measures in three endemic communities of Mayuge District, Uganda using rapid ethnographic appraisal methods.

The findings indicate that people mainly think schistosomiasis is caused by drinking unboiled lake-water, which resulted in doing nothing to reduce skin contact with lake-water. The schistosomiasis symptom of a swollen stomach was interpreted as 'ekidada', a local disease caused by witchcraft, which resulted in not taking praziquantel, and using local herbs instead. Others did not like praziquantel because of its side effects including diarrhoea, and abdominal-pain and ideas that it reduces fertility and life-span. Open defecation was a common practice across all communities. Generally, schistosomiasis was considered hard to prevent since the lake is the major source of water, and social-economic livelihood. All these perceptions negatively affected schistosomiasis control measures. We, therefore, advocate for a contextualized health-education programme, alongside improved access to safe-water and latrines.

## Introduction

Schistosomiasis commonly known as bilharzia is a neglected tropical disease (NTD) [1,2] affecting over 230 million people worldwide [3,4]. Nearly 780 million people are estimated to live at risk of infection across 76 countries [5,6], of whom 660 million live in Africa [7].

Human schistosomiasis is acquired when people contact fresh water that contains infective larvae, called cercariae, which actively penetrate the skin, develop into adult worms, pair up, and sexually reproduce eggs which are then excreted in human urine or faeces (depending on species). In freshwater, eggs hatch into miracidia which penetrate intermediate snail hosts where they asexually reproduce, releasing thousands of free-swimming cercariae back into the water [8–10].

Schistosomiasis causes a range of symptoms from anaemia, abdominal pain, stunted growth and reduced cognitive development in children [11], leading to more severe illness and up to 200,000 deaths per year globally [12]. The World Health Organization (WHO) recommends praziquantel mass drug administration (MDA) to reduce morbidity, and ultimately transmission [8] aiming for the elimination of schistosomiasis as a public health problem by 2030 [3]. Praziquantel kills many of the adult worms but does not prevent re-infection and does not kill juvenile worms [13,14].

In Uganda, *Schistosoma mansoni*, causing intestinal schistosomiasis, is a significant problem: 16.7 million Ugandans live at risk of infection [15] with over 4 million people, across over 80 of Uganda's 134 districts, infected [16]. Control in Uganda began in the early 1990s', targeting morbidity with drug treatments [17]. In 2003, annual MDA [18] coupled with health education, started to be rolled out, this was scaled up in 2004 as a national MDA programme following WHO guidelines [19]. By 2006, the prevalence and intensity of infection had been greatly reduced in many treated communities [20]. However, more recent estimates have shown that infection may have increased again, from 9.1% estimated as the national prevalence in 2012 [21], to 22.1% in 2016 and 29.0% in 2017 with children aged 2–4 years having the highest prevalence for schistosomiasis of 36.1% [22,23]. This reported increase [22], may be due to improved diagnostics and mapping [24]. However, also, population growth and human movement [25] without improvements in sanitation infrastructure [26]. These, alongside limited understanding of transmission and the ability to change high-risk behaviour likely all lead to increased, or at least maintained, transmission levels, despite MDA.

Schistosomiasis infections are particularly high in settings with poor sanitation and limited safe-water infrastructure and access [27]. Communities along the shores of Lake Victoria, the largest lake in Africa, are highly endemic with >50% of school-aged children infected [28,29]. In Mayuge District, a lakeshore district with extensive shoreline and some islands, schistosomiasis is the third-ranked disease after malaria and pneumonia [30] with community prevalence estimates of up to 75%-80% [4,16,31] and 95% in school-aged children [32].

Despite the recommended annual community-wide MDA in high endemicity areas [4,24,33], a survey in Bugoto, a fishing village in Mayuge and one of the three communities studied here, recorded MDA coverage of only 46.5% [15]. 35.3% of the population eligible to take praziquantel reported never to have taken it, despite over 10 years of MDA [15]. This significantly reduces the MDA impact on host morbidity and transmission [34]. Reasons for not taking praziquantel included not knowing about MDA, thinking MDA was only for children, being away during MDA, and fear of side effects [15]. The WHO recommend a minimum coverage of MDA of 75% to reduce morbidity and transmission [24,33,35]; this is far higher than the achieved in this high endemicity community [15] and many others [28,36]. Despite continued high endemicity in Mayuge [37,38], across Uganda [39] and sub-Saharan Africa [40,41] our understanding of why some individuals and groups are not taking MDA, or understanding how to, or manage to, change their high-risk behaviour, is still lacking.

In Uganda, one study in Koome Islands, Lake Victoria, found that community members had heard about schistosomiasis from health workers, community leaders, and schools, but had gaps in knowledge of how you catch, transmit and prevent infection [42]. Another study indicates that schistosomiasis has persisted in communities with diverse exposure patterns

having high rates of re-infection [43]. In Kenya, 55% of people in a *S. mansoni* endemic community had limited understanding of how schistosomiasis was caught, transmitted and prevented [12], and in Nigeria, 38.6% of individuals in endemic areas did not know any symptoms, 67% had no idea about how it was transmitted and 63.8% did not know how to avoid infection [44]. People perceived that schistosomiasis is caused by salty or sour food and transmitted through sharing latrines [44]. In Tanzania, people's beliefs about supernatural causes of schistosomiasis resulted in traditional healers being consulted in favour of praziquantel treatment [45]. Participants perceived schistosomiasis to be hard to prevent, and the lack of specific severe symptoms negatively influenced whether action was taken [45]. All these studies indicate the existence of varied perceptions of schistosomiasis in endemic communities and the overall low levels of understanding of water contact as a critical factor, despite a wide range of perceptions and practices.

In Mayuge, high infection levels, rapid re-infection and ongoing transmission still exist despite over 15 years of MDA [10,37], and praziquantel coverage remains low [15]. Our aim was to provide a deeper understanding of how different perceptions about schistosomiasis and its transmission shaped people's choices on prevention and treatment practices and how they acted on those choices. We used rapid ethnographic appraisals (REAs) [46–49] to triangulate observations with self-reported actions in different settings (for example interviews and focus group discussions), thus more clearly identifying the relationships between the practices people report, the settings in which they report them and their observed actions. We explored: 1) people's understanding of schistosomiasis symptoms; 2) how perceptions of schistosomiasis affect treatment uptake; 3) people's perceptions of how you contract the infection; 4) how they think a person can transmit infection onwards and responsibility for this, and 5) perceptions of prevention and control measures.

## Methods

### Ethical approvals

Ethical clearance was obtained from the University of Glasgow's College of Social Science Ethics Committee (CSSEC 400160134), Uganda Virus Research Institute Ethics Committee (GC/127/20/06/601) and the Uganda National Council for Science and Technology (UNCST) (SS 4241).

Informed consent was sought from all participants before participation in focus group discussions or in-depth interviews. Study objectives were explained in Lusoga, a commonly spoken language in the studied communities. Informed consent documents including the participant information sheet and consent forms were in Lusoga. For the participants who were less literate the research assistants read out the study information providing ample time for questions and a two-way dialogue to ensure participants comprehended information before the decision to participate was taken. Individuals willing to participate signed or placed a thumbprint on a written consent form prior to enrolling in the study and were aware that they could withdraw at any time with no effect on their subsequent schistosomiasis treatment or legal rights. Verbal consent, where possible, was obtained during participant observations. For children, parents or guardians provided informed (signed or thumb printed) consent, and signed or thumb printed assent was obtained from all children aged eight years and above. All photographs were taken with verbal consent.

### Study area

The study was performed in three communities highly endemic for *S. mansoni*: Bugoto, Bwondha, and Musubi, all on the shores of Lake Victoria in Mayuge District, Eastern Uganda

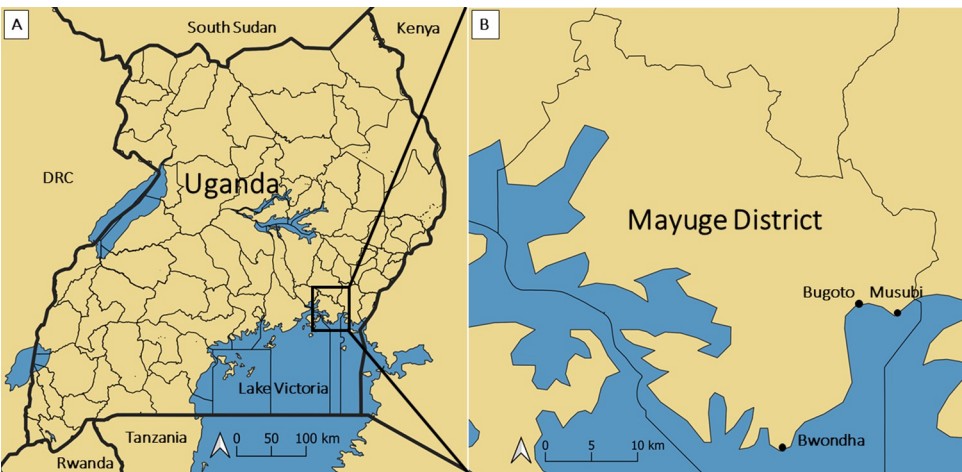

**Fig 1.** (A) Uganda map showing Mayuge District. (B) Mayuge District map showing the three study site villages: Bugoto, Bwondha and Musubi. Created using QGIS 3.14 Software (QGIS Association. QGIS Geographic Information System. 3.14 ed2020) using a base layer from Natural Earth (http://www.Naturalearthdata.com) for the reference maps, with district boundaries created using Uganda Bureau of Statistics (Uganda Bureau of Statistics (UBOS). Uganda Administrative Boundaries GIS Database. Kampala, Uganda: Government of Uganda, UBOS; 2006) [51].

(Fig 1). In these communities prevalence and intensities are high, for example 71% of people are infected in Bugoto community, [50] with likely similar infection levels in Musubi and Bwondha [27,37]. Infection risk behaviour is also high [50], whilst MDA coverage remains low [15,50] despite over 15 years of annual MDA indicating that additional, locally informed, improved interventions are needed.

Fishing and agriculture are the major economic activities and the lake is the major source of social and economic livelihood. These communities are predominantly populated by people who speak Lusoga, one of the languages spoken in the eastern part of Uganda. Bwondha is the largest of the three study communities, with an estimated population of >60,000, compared with 6,000 in Bugoto and 2,000 in Musubi [51]. Bwondha's main economic activities are fishing, fish processing and trade followed by agriculture and small businesses including shops, kiosks, eating places and bars. Bwondha is a hub with many boats transporting people to islands and a market selling agricultural produce from the mainland and islands. In Bugoto and Musubi people mainly practice fishing, and agriculture including rice farming in the swamps or at the lakeshore. All three communities are close to the lake with several houses <30 metres from the shore. People have high rates of mobility, particularly fishermen travelling to higher fish stocks across these communities and/or neighbouring islands. The main water source is the lake, used for washing, cooking, bathing, swimming and other domestic, financial and social activities. Interactions with lake and swamp waters expose people to a high risk of catching *S. mansoni*.

## Study design

Fieldwork occurred between September 2017 and April 2018. This was a qualitative study using REA [46–49] to explore individually and socially constructed ideas about schistosomiasis symptoms, transmission and control, comprising interviews and focus group discussions together with transect walks and participant observations, and structured and unstructured observations of everyday forms of interaction with water, sanitation and hygiene (WASH).

## Data collection

One female (EN) and one male (LM) Ugandan social science researchers, fluent in Lusoga, but not from the communities themselves, spent six weeks in each of the three communities. After introductions to the District Health Officials, Village Chairpersons, and people in the communities, Village Health Team members (VHTs) guided EN and LM on transect walks across their communities, highlighting key WASH sites. Structured observations were then undertaken at all types of water sources: lake, water taps, boreholes, swamps, and wells. Each observation period lasted 20–60 minutes and were undertaken in each community in mornings, afternoons, and evenings, on different days of the week, to maximise observations and our understanding of how behaviours may vary across time, days and villages. Using a REA approach, all of these methods were underpinned by a commitment to immersion and participant observation. EN and LM participated in everyday activities including attending mosque services, funerals, football matches, market days, MDA administration, village meetings, drinking in bars and eating in public eating places. This enabled the researchers to obtain people's views about schistosomiasis in the context of what they discuss in everyday life. There were opportunities for chatting informally with people in the village in general as well as for those the research team aimed to get more detailed information from in the interviews and discussion groups.

Community participants for the in-depth interviews and the focus group discussions were purposively selected to provide community-wide insights. In each village, we began with an introductory focus group discussion with 8–10 key figures in the village such as the teachers, community leaders, VHTs, health workers, and beach management leaders. Four subsequent focus group discussions were conducted in each village, each with 10–12 people, of either older (>35 years of age) men, younger (18–35) men, older women, and younger women. Twenty in-depth interviews with fisherfolk, farmers, car washers, washerwomen, traditional healers, and business people were then undertaken to probe more deeply into issues raised during the focus groups. Four focus group discussions were also conducted with children in Bugoto and Musubi, comprised of children both in and out of school (one with children aged 8–11 and one with 12–14 years in each of the two communities). These group discussions focused on children's understanding of schistosomiasis and its control.

## Data management and analyses

Data comprised of audio files of interviews and focus group discussions, plus observation notes and daily research diaries. Notes and diaries were typed up and uploaded, along with the recordings, to secure password-protected computers each day, and stored in a locked room throughout data collection. Data were transferred to the secure Medical Research Council Unit server in Entebbe. All audio files were transcribed, translated and anonymized by EN and LM before being shared with the wider research team.

These documents were coded by EN, LM, LP, and KM on Nvivo 11. Coding was thematic [52], with an initial coding frame developed by LP and KM; the analysis was semi-inductive with new codes being added to reflect emergent themes as coding progressed. Coded data were analysed using iterative categorisation [53]. Quotes were analysed against the five main objectives, each including sub-categories, but based around people's perception of schistosomiasis: 1) symptoms; 2) praziquantel treatment; 3) contracting the disease/infection; 4) transmitting the infection onwards, and 5) control methods to reduce contracting and/or transmitting the infection.

**Table 1. Social demographic characteristics of study participants for the in-depth interviews.**

| Characteristics | In-depth Interviews (N = 60) |
|---|---|
| **Sex** | |
| Female | 27 |
| Male | 33 |
| **Age** | |
| 18–35 | 30 |
| 36–45 | 13 |
| 46–55 | 13 |
| 56+ | 4 |
| **Occupation** | |
| Fishing (Inc. fish processing, fish drying, fish smoking, fish trading) | 25 |
| Farming | 10 |
| Potter | 1 |
| Village Health Team member | 7 |
| Others (Business, community and religious leaders, teachers and security officers) | 14 |

## Results

Between September 2017 and April 2018 qualitative data were collected from the three *S. mansoni* endemic communities of Bugoto, Bwondha and Musubi. We conducted 26 transect walks, 34 observations, 24 participant observations, 19 focus group discussions with both children and adults and 60 in-depth interviews (for interview participant characteristics see Table 1).

### General observations

Observations and transect walks revealed that fishing and agriculture were the major economic activities. All three communities have extensive Lake Victoria shorelines. Almost all members of the community interacted with lake water, for different reasons and at different times of the day, with some coming into contact with lake water more than twice a day. The lake was the major source of livelihoods and main water source for washing clothes, utensils, motorcycles and bicycles, and for bathing, cooking, cleaning, and other household activities as well as drinking (Fig 2). Some of these activities including washing, bathing and swimming were performed at the lake or lakeshore, whilst for other activities people collected water from the lake for use elsewhere.

The houses in the three communities were generally crowded with few latrines available relative to the population size. Existing latrines were of poor structure (Fig 3), many were temporary due to high water tables, they filled up quickly and when it rained the faecal matter was washed to the lake. Musubi had no public latrines; in Bugoto and Bwondha the public latrines were in a poor state, people defecated and urinated missing the holes and/or went behind the latrine buildings. Several houses did not have private latrines, especially the rental houses. During transect walks in the communities, lakeshore, and fields, we observed a lot of open defecation. When it was raining, due to the ground sloping towards the lake, rainwater, likely containing faeces, flowed from up to 3 kilometres away to the lake.

**1. Perceptions about symptoms of schistosomiasis.** From our informal discussions with community members, schistosomiasis was considered a big problem in all of the communities. The most mentioned symptom of schistosomiasis, regardless of community, age, gender and level of education was a swelling of the stomach (see Fig 4).

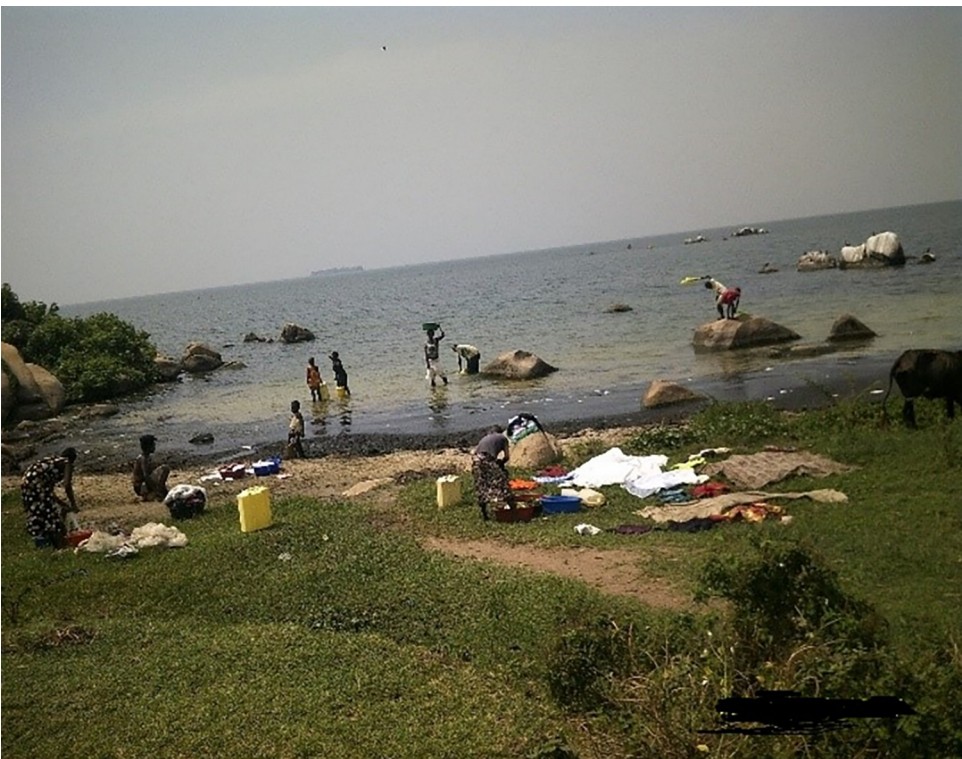

**Fig 2. Common activities at the lake.** Additional to the lake, in all three communities we observed taps with clean filtered lake water and boreholes drilled to provide naturally occurring water from the ground. However, these sources were not reliable as they were often locked, out of water, and/or faulty, especially in Bugoto and Musubi. Tap water was pumped from the lake by a solar pump which could not function during cloudy days and rainy periods. Even the functioning boreholes were often not used because people said the water was salty. In Bwondha there were several shallow wells with big queues. In Musubi, we saw people getting water from swamps and wells which were also used by animals to drink water.

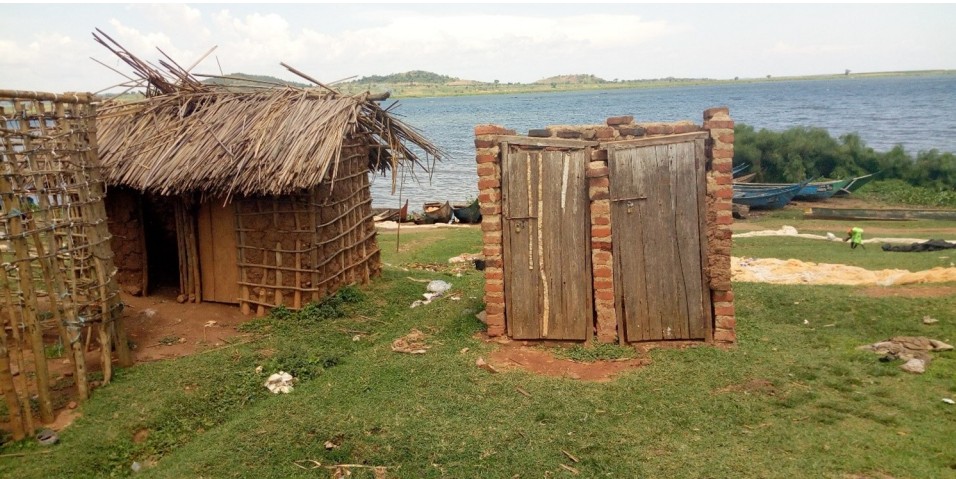

**Fig 3. Examples of pit latrines in the studied communities.**

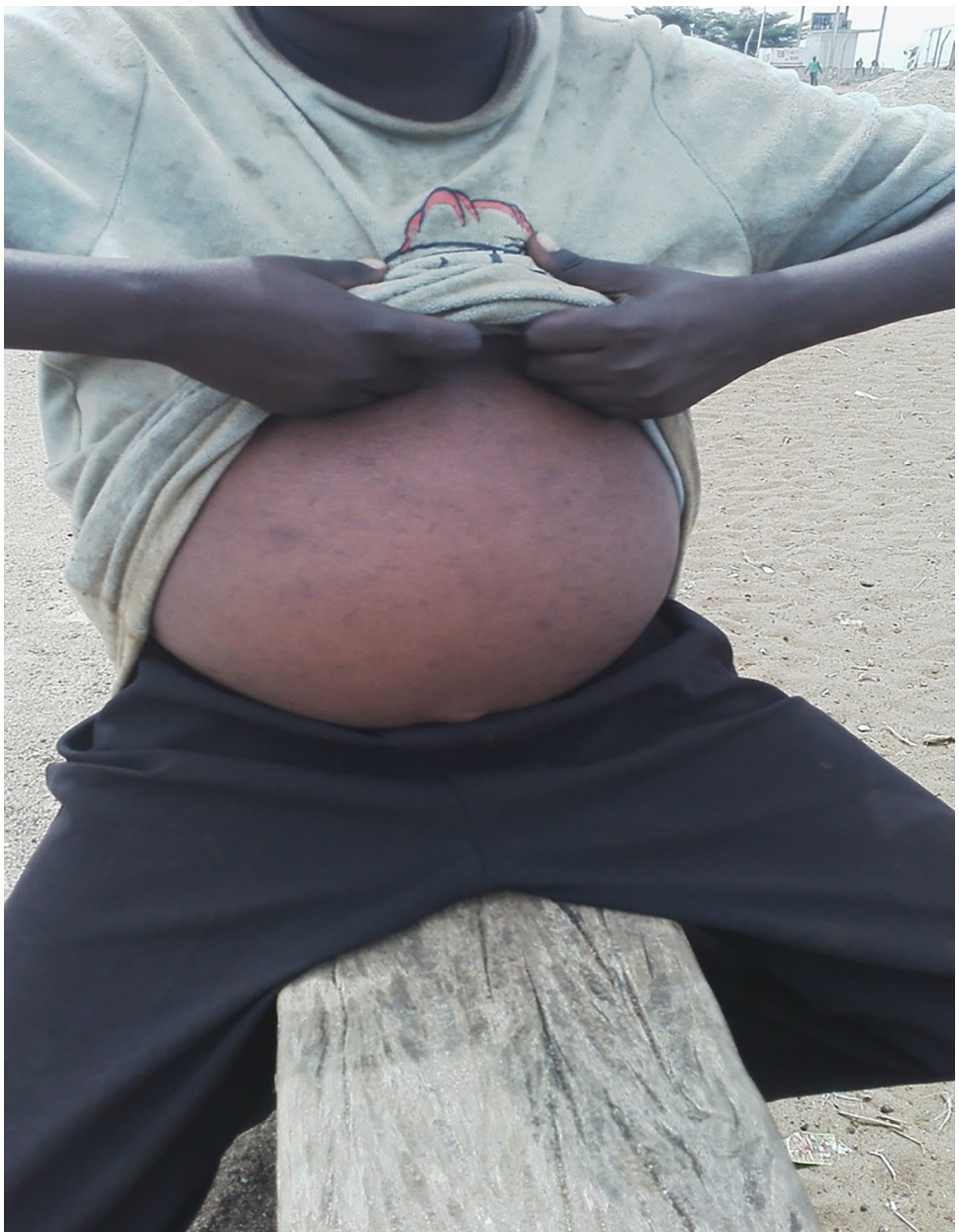

**Fig 4.** A 13-year-old boy with an extended stomach. Some community members, including some with enlarged stomachs, interpreted swollen stomachs as a symptom of ekidada (a local disease that was considered to be a result of witchcraft).

*"Personally whoever I see with a swollen stomach I know he has bilharzia." (Community leaders FGD Bwondha).*

*"Even if that person has not gone for a check-up, for us we know that whoever has a swollen stomach has bilharzia as long as he is not bewitched [ekidada]." (Older men FGD Musubi).*

*"It seems the person who bewitched me also added charms of not getting healed, that is why you see up to now I look like this as you are seeing (swollen stomach). The witch doctor I went to in Bukhooli made me defecate a toad frog but the stomach is still swelling." (55-year-old male fishermen IDI Musubi).*

Even some of those who said a swollen stomach was a sign of schistosomiasis reported that previously they also used to think that the swelling was due to witchcraft. However, some community members, including VHTs, reported that their perception changed after receiving health education and sensitisation.

*"I got to know about bilharzia through the bilharzia workshops I have attended for years and years by health workers who are in charge of bilharzia, that is when I got to know it is true there are parasites that make the stomach swell, but not* ekidada, *that is bewitched." (46-year-old male VHT IDI Bwondha).*

However, there were also other ways in which a large stomach was interpreted, such as a sign of being wealthy.

*"But there are men who have a big stomach, they claim that they have money." (Young women FGD Bwondhai).*

*"Me I know bilharzia it is something that turns one into another thing, and someone will be labelled rich because of a big belly yet it is not richness but illness." (Young men FGD Bwondha).*

Some respondents interpreted swollen legs as a symptom of *ekidada*, schistosomiasis or elephantiasis (lymphatic filariasis), linking two symptoms (a swollen stomach and swollen legs) with two different diseases (schistosomiasis and elephantiasis). This came out most clearly in the community that had a man with elephantiasis (with swollen legs), and people asked about the symptoms of schistosomiasis often referred to him in their answers.

*"I have my brother, his feet swell and people say that he acquired bilharzia from the lake that made the feet swell. . ." (Young men FGD Bugoto).*

A few people mentioned weight loss amongst those with extended stomachs as a symptom of schistosomiasis, but this was more commonly related to HIV, further demonstrating the multiple ways in which individual symptoms can be interpreted, including in line with other diseases found in the community.

*"For the adult to get to know that he has bilharzia. . ., you get to know from the stomach, the stomach swells, sometimes the limbs swell. Some people even say they have HIV but when they have bilharzia. . . They become slim when it is only the stomach that is big on them." (40-year-old female silverfish trader IDI Bugoto).*

Some people in the community could not describe symptoms of schistosomiasis, saying a medical test is needed.

*"We can tell the symptoms after doing a medical check-up because someone can have a swollen stomach when he is not sick." (Community leaders FGD Bwondha).*

Other symptoms of schistosomiasis that were occasionally mentioned included diarrhoea, abdominal pain, anaemia, vomiting, loss of energy, body itching, becoming pale, yellowish skin/eyes, big/sunken cheeks, loss of appetite and getting sick with malaria often.

*"Some present with signs like anaemia, becoming pale and general body weakness." Young men FGD Bugoto).*

*"For children who play from the lake and also swim it is easy to know because they always complain of abdominal pain, general malaise and diarrhoea." (Younger women FGD Musubi).*

**2. Perceptions about the treatment of schistosomiasis.** The majority of participants across all communities reported that schistosomiasis can be treated by taking praziquantel that is administered during the annual MDA.

*"Me I was an original fisherman and am still a fisherman, I remember my time to swallow those tablets (praziquantel)... by then I had a swollen stomach as if I am rich yet I am poor, but after two years when I took the tablets there was some change... so bilharzia can be treated." (Community leaders FGD Musubi).*

However, despite mentioning awareness of praziquantel to treat schistosomiasis, some expressed negative attitudes towards it. A large number of children and adults did not like praziquantel because of side effects like vomiting, diarrhoea, abdominal pain, nausea, dizziness, making them weak and unable to go to work, especially the fishermen who did not want to miss fishing after taking praziquantel. Some people reported having experienced side effects, whilst others only heard rumours about side effects from the community or friends and decided not to take/retake the tablets. Several community members were heard saying they will never swallow praziquantel again.

*"He told me, while in Bwondha he took bilharzia drugs but he got diarrhoea and didn't go to work. That he just put the mat near the latrine so that he could run to the latrine at any time, he became weak after and had to go for further treatment, he said ever since that time he will never take bilharzia drugs again." (Participant observation in a bar with a young fisherman in late 20s in Bugoto).*

In addition, some adult men and women did not like the treatment because they thought praziquantel affected fertility and reduced life span.

*"Some people say they reduce sexual manpower. Some people say that those tablets are like a method of family planning, that when you swallow them a lot you don't produce." (40-year-old female silverfish trader IDI Bugoto).*

In addition, others across all communities perceived praziquantel treatment not to be effective because some people still had extended stomachs after treatment. They said praziquantel only gave temporary relief and therefore wondered why they were required to swallow the tablets every year.

*"Me I think the tablets they give to people are not important, there is a child, they have given him those tablets but the stomach failed to reduce, it is still swollen, they have given him*

*tablets several times, whenever they come they give him but the tablets failed to respond to the illness. So I think those tablets don't work."* (Older women FGD Bwondha).

Furthermore, some religious groups did not allow followers to swallow tablets. They believed that they are '*satanic*' and even when they fall sick they do not seek care from hospitals or health centres.

*"Apart from ignorance, there is a religion that does not allow such things [MDA], in the religion of 'Ngiri Nkalu' people don't swallow tablets, they are not injected, they don't educate their children, when we are immunizing for polio we just fight with them and take them to police."* (46-year-old male VHT IDI Bwondha).

Participants who mentioned a swollen stomach in relation to '*ekidada*' (caused by witchcraft) sought herbs from traditional healers. In several instances, this was after taking praziquantel and seeing no improvements.

*"I thought I was suffering from bilharzia because I had spent a long time fishing, I went to the VHT and she gave me tablets (praziquantel) but it did not respond. Then people told me I was suffering from* ekidada *that needed local herbs to get healed."* (55-year-old male fisherman IDI Musubi).

Some people were reported to have never swallowed praziquantel because they do not present with any symptoms, they felt healthy and therefore saw no need to take the treatment.

*"The VHTs move door to door giving out tablet[s] but some people refuse to swallow, they tell them, for me, I don't have bilharzia as if they have ever tested."* (30-year-old female fish smoking IDI Bwondha).

**3. Perceptions about how schistosomiasis is contracted.**   Participants had varied perceptions about how people contracted schistosomiasis. Across all the three communities, the majority of participants, irrespective of gender, age and social status said people contracted schistosomiasis through drinking unboiled lake water.

*"Adults catch bilharzia through drinking unboiled water, and even if you boil the water and use a wet mug to drink boiled water you still catch bilharzia. That water mixes with the unboiled."* (50-year-old female silverfish trader and community leader IDI Bwondha).

*"Children swallow that water while swimming."* (Children FGD 8–11 years Bugoto).

As a result, many participants did not mention any measures about reducing skin contact with lake water. Only occasionally was infection associated with water contact as a result of bathing, washing, swimming or when people stand, step or walk in swamps, stagnant and/or running water.

*"Someone that goes to the lake might contract bilharzia from the water, and also one that normally gets in contact with dirty water for example those that are in rice farming in the wetlands, also if they go into the lake, you might contract bilharzia."* (31-year-old female silverfish trader IDI Bugoto).

However, even though some respondents mentioned knowing this route of infection, they reported not avoiding the lake because it is the only source of free water and livelihoods.

*"We cannot avoid catching bilharzia because it is found in the lake where we work from and that is where we get water from." (Older men FGD Bwondha).*

The other frequently mentioned cause of schistosomiasis was through urinating in swamps and lakes or through open defecation. Some people thought if they defecated or stepped where a person with schistosomiasis defecated or urinated they could also catch it.

*"You may defecate where they have already defecated and the parasites around may enter you." (30-year-old female fish smoking IDI Bwondha).*

*"Children who step where they have defecated also catch bilharzia." (Children FGD 8–11 years Bugoto).*

For some this related to stepping in faeces anywhere, or stepping barefoot in areas where people throw rubbish, while for others, the risks more specifically related to stepping in faeces around the hole of the pit latrine, stepping in urine on the floor of the latrine, stepping in water used to clean the latrines, or entering the bathroom barefooted.

*"You catch bilharzia if you walk barefooted in dirty places. Like now, this is a rainy season but somebody may decide to walk barefoot yet the place is dirty and it has cow dung, so this exposes one to bilharzia." (33-year-old female fish smoking IDI Bwondha).*

Another described route of contracting schistosomiasis was through eating raw, dirty, uncovered, half-cooked or contaminated food.

**4. Perceptions about schistosomiasis onwards transmission and responsibility of transmission.** The way people thought schistosomiasis was transmitted onwards from humans varied across groups, with the most common route being through open defecation or urinating in the lake. Across all three communities, all categories of participants, including fishermen themselves, considered fishermen as the main group that practice open defecation. As a result, most participants attributed transmission of schistosomiasis to fishermen.

*"Interviewer: Who are those you said cause bilharzia?*

*Respondent: They are the fishermen. They are the ones that usually misuse the lake by having to urinate in it, once nature calls they can't urinate anywhere else but rather in the lake...They also defecate in there, they will do like this while on boats on the lake (demonstrated how fishermen bend while on the boats to defecate) and defecate in the lake."* *(61-year-old male community leader IDI Bugoto).*

*"With the majority of fishermen, the lake is their latrine. Even when they feel like defecating while here at the landing site; they will hold up the faeces until they go to the lake." 37-year-old Female silverfish trader IDI Bugoto).*

The fishermen perceived themselves to be responsible for transmission, describing practicing open defecation (Fig 5) as the norm and a habit that has existed since their forefathers. They considered the lake as a natural latrine where they had free water for cleansing after defecating, the waves removed faeces from the lakeshore, and/or the lake was a big place that cannot be changed by a few faeces that are dissolved immediately without affecting the lake.

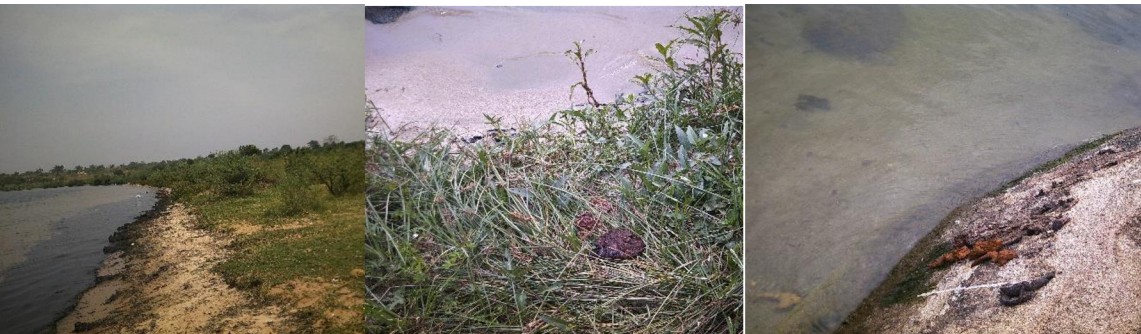

**Fig 5. Open defecation on rocks and bushes at the lake shoreline.**

*"On that issue of the latrine, you have asked why the fishermen leave home latrines and poo at the lake but the issue is, people who work at the lake are mature and they go at the lake for work, if they feel like defecating they can't go back home when at the lake there is a natural latrine.*

*Interviewer: What is that natural latrine?*

*(All respondents laughs) The lake." (Community leaders FGD Musubi).*

Some fishermen reported performing open defecation because of limited access to latrines at home or their rental houses, a lack of public latrines, and an inability to meet costs of public latrines, whereas it is free to defecate at the lake.

*"You see health workers, I wouldn't want to lie to you, am a fisherman and I do not have a latrine, so if I go fishing and feel like pooing, I end up defecating in water because I do not have an option, am among those who do open defecation." (21-year-old male fisherman observation Musubi).*

*"We cannot avoid defecating in the lake because we are always at the lake the whole day. It is unavoidable." (Younger men FGD Bwondha).*

Some fishermen reported using faeces to attract fish.

*"For me by the time I started fishing, we even used to buy faeces to use in trapping fish, after putting the fishing net you go where the wind is coming from and defecate there so that the faeces flows towards the net, when the faeces join the fishing net the fish that were eating the faeces also would like to continue eating and they enter the net." (73-year-old male fisherman IDI Bugoto).*

Another group described as engaging in open defecation practices were members of particular, usually other, tribes. This also tended to be attributed to women, on the basis that they believed that if they used latrines they would then not be able to produce children or keep them safe.

*"Majority people especially the Samia tribe don't have latrines, they don't go to latrines apart from the men, women from the Samia tribe if they defecate in a latrine they believe they are throwing their babies in the latrine." (73-year-old male fisherman IDI Bugoto).*

There were a few people who thought schistosomiasis was transmitted by flies, or through shaking hands and sharing food. Some people suggested isolating those with schistosomiasis and not sharing anything with them to avoid transmission.

*"A person with bilharzia can tell a friend not to eat on the potato he ate on, they should tell friends, go and have yours, don't drink on my water, don't drink on mine. You don't share anything with a person with bilharzia." (53-year-old male fisherman IDI Bugoto).*

## 5. Perceptions about how people can prevent infection and onward transmission

Across all communities, schistosomiasis was perceived as hard to prevent because people have to use the lake for daily social and economic livelihoods (see Fig 6).

*"To me, at this landing site, there is no way we can prevent bilharzia because you may tell me not to step in water but I have to fetch water from the lake where the parasites are." (Community leaders FGD Bwondha).*

*"If it is like that, then we cannot avoid catching bilharzia because for us we take lake water and we may catch bilharzia, we work from the lake and we cannot avoid the lake because that's where we work from, get water for drinking, bathing and washing and sometimes we bathe from the lake." (21-year-old male fisherman observation Musubi).*

Some people described a range of actions aimed at reducing schistosomiasis risk, with most reporting boiling drinking water. However, during focus group discussions, other participants mentioned that many of those who reported boiling water, do not actually do it, and reported that the majority of people drink lake water in its raw form, as fetched from the lake.

A few participants mentioned fetching drinking water from boreholes, shallow wells or taps to avoid drinking lake water. However, others reported continuing to use lake water for

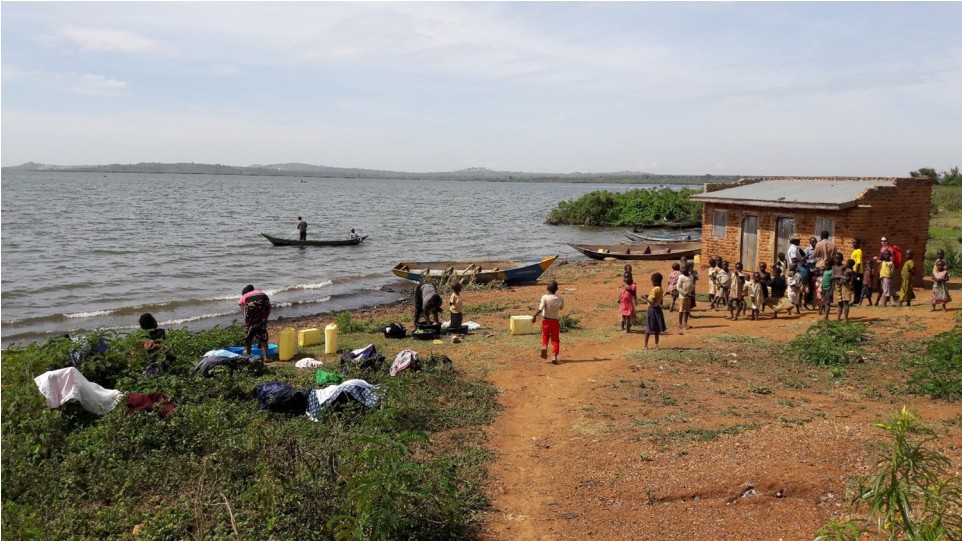

**Fig 6. Social activities performed at the lake.**

drinking because other water was expensive. A 20-litre jerrycan of water in Bugoto was between 500–1000 Ugandan shillings (10 to 20 British pence) and 200 Ugandan shillings (four British pence) at a tap/shallow well in Musubi and Bwondha respectively.

> *"We draw from the lake because the drinking water that they bring in the community is just sold, so those without money mainly get from the lake. . . They sell at 700 shillings. People don't afford to buy that water, that's why they get drinking water from the lake". (47-year-old fisherman Bugoto).*

Some mothers reported bathing babies in boiled water as a way of protecting them against schistosomiasis (a recommended safe strategy) and keeping the umbilical cord clean; this is mainly practiced until the umbilical cord falls, after approximately two weeks. However, adults did not boil water to bathe older children or themselves, unless it was very cold.

> *"For us, we don't fear the parasites, our bodies are already strong. Our bodies got strong long ago." (30-year-old female fish smoking IDI Bwondha).*

Preventive measures like boiling were perceived by some men to be expensive.

> *"What makes us not boil drinking water is that firewood is expensive, one piece is being sold at 300 shillings or 1000 shillings, so people say they can't waste the firewood or charcoal on boiling water yet they have to cook food." (Older men FGD Bugoto).*

Some people, especially women, considered leaving water in the sun, boiling, or filtering it (all methods suitable for making it safe to bathe in) as a luxury and waste of time.

> *"People think about the time and charcoal wasted in boiling water, they even say, 'how can I boil water just for washing. I can't boil'. . . . Also, people don't have that time to first leave it in sun, someone wants to cook food and she has no time to leave water in the sun first, they say, 'aah! If am to get infected let me be infected, I will go to the hospital." (26-year-old female silverfish trader IDI Bugoto).*
>
> *"I will not lie to you that I can waste time to filter water for bathing, that's haram (forbidden). I get from the lake, pour in a trough and bathe." (Young men FGD Musubi).*

Those who mentioned leaving water in the sun did it unintentionally, whilst others mentioned the risks of leaving the water in the sun.

> *"The sun just finds the jerry can there, it is not done intentionally that we want to leave water in the sun to warm it, the sun just finds water there." (55-year-old male fisherman Musubi).*
>
> *"And if I find a child leaving water in the sun I will even cane him because the sun destroys the jerry cans. . . . . ., if you leave in the sun a cow can step on it, If a thief looks each side when there is no one seeing him he can just put the jerry can on his head and walks away as if it is his." 53-year-old male fisherman IDI Bugoto).*

Linked to this, some people said that leaving water in the sun could bring misfortune. They were told this by their forefathers and therefore just continued practicing that. However, some people reported this practice to be reducing amongst those who embraced Christian and Islamic religions and that it is still only being practiced by a few.

*"We fear to leave water for bathing in the sun because we were told that when you leave water in the sun, it is ghosts of your ancestors that bathe that water first and for you when you bathe on that water, you shower yourself with bad omen. That's why we fear to leave water for bathing in the sun."* (Young men FGD Musubi).

## Discussion

In this study, we used REA to explore how people perceive *S. mansoni* symptoms, treatment, becoming infected, onwards transmission, and the difficulties of prevention based on measures currently available, across three highly endemic communities that have received praziquantel MDA for over 15 years. We focused on the information that people had about schistosomiasis as well as the ways they reacted towards it for the control of schistosomiasis. We mainly found that what people thought was the cause and transmission of schistosomiasis shaped what they did to treat infections and symptoms and to control their risk of infection or onward transmission.

Some participants described symptoms such as swollen legs as being linked with schistosomiasis. This is a symptom of another NTD, lymphatic filariasis (elephantiasis), also found in the area and also treated at a community level with MDA. This often led to confusion in understanding symptoms, where people interpreted symptoms associated with one disease with others, or interpreted a single symptom into multiple causes. Symptom confusion may be due to mixed messages around NTDs and other diseases in general [54], as well as mixed messages from radio announcements mentioning that schistosomiasis causes body swellings such as of the stomach and legs [55], with the latter actually associated with elephantiasis. Many symptoms caused by schistosomiasis and other diseases are often non-specific and can have multiple causes, further compounding the problem.

The way our participants understand schistosomiasis, based on interpretation of its symptoms, is in line with existing findings about how people come to understand diseases in relation to other local diseases [56,57]. For instance, some participants related a swollen stomach, a common sign of severe schistosomiasis morbidity, to be a result of being bewitched. Elsewhere, such as malaria in Ethiopia, HIV in Tanzania and Haiti, disease symptoms were also associated to witchcraft/magic [56–58], further highlighting the often blurred lines between infectious diseases and magic/witchcraft and subsequently biomedicine versus traditional healing.

Furthermore, interpreting a swollen stomach as witchcraft, treated by herbs, is in line with other research on schistosomiasis in Uganda and elsewhere in Kenya, Tanzania, and Nigeria, where it is also treated by herbs and/or spiritual healing from traditional healers or witch doctors [44,45,55,12,59]. Participants understanding that a swollen stomach was a result of witchcraft or bilharzia, may also be linked with inadequate information messaging [55]. District officials, health workers and VHTs, as well as some NGO radio campaigns, use the term *ekidada* when communicating about schistosomiasis, without realising the additional 'bewitched' meaning in these communities [55]. Our findings also show that this mixed interpretation of a swollen stomach potentially limited praziquantel uptake, with herbs from traditional healers or witch doctors used instead or when praziquantel was seen to have been previously ineffective.

We report that some people did not like praziquantel because of its side effects, and others abandoned it in favour of herbs at a cost. This supports another qualitative study in Lake Victoria communities [17] and a quantitative [15] study previously performed in Bugoto. Our participants especially the fishermen, reported missing praziquantel because it was

administered in the afternoon when they are about to go to work and a fear of it making them weak and missing fishing whilst others feel healthy. This adds to the complexity surrounding the fear of side effects and non-adherence. This overlaps with research from Zanzibar where people reported missing praziquantel because they felt healthy and were busy [60]. These complex interacting bio-social factors all contribute to the low drug coverages of 46.5% seen in these areas [15], far below the WHO target of 75% praziquantel treatment coverage [4,54].

The repeated low praziquantel coverage may in part explain the continued high infection intensities seen across these communities [37]. However, high intensities are also strongly associated with high levels of water contact activities [29,61,62]. Corroborating quantitative research in Bugoto [50] that showed people contacting lake water daily and for long hours, we observed people engaging daily with high-risk water contact activities, regardless of age and gender, reinforcing the notion that schistosomiasis is hard to prevent. We observed and heard that people continue with their daily activities at, and in, the lake whether they are aware of the risk or not, because it is the major source of livelihood and free water. The lack of reliable alternative water sources resulted in people interacting with lake water even when they wanted to avoid it. We observed that the lack of safe water is a communal problem leading to frequent lake contact, including being a source of play for children. Even though some people do not drink from the lake, they depend on lake water for all other water activities. Research in Western Uganda also shows that schistosomiasis was considered hard to prevent because the lake is the only source of free water [63] and many other Ugandan studies highlight a lack of safe water in fishing communities [49,63–65]. The taps available in Musubi and Bugoto villages were unreliable, they frequently broke down, some were faulty or locked and on cloudy or rainy days the taps run out of water because they used a solar pump. The boreholes were also faulty, and the few functioning ones produce salty water that people do not like. This dislike of hard and salty borehole water, and a preference for lake water, has also been reported in other Ugandan fishing communities [64]. Thus, the absence of acceptable safe water leads to continued contact with potentially infected lake water, culminating in rapid infection and re-infection, including post treatment [10,31,66].

From the REA, we also confirmed that latrine coverage was low, contributing to open defecation, supporting other findings in the area [15]. According to Mayuge government data, latrine coverage in Mayuge is 62% [30], however other studies report much lower levels of only 33% of people living by Lake Victoria in Uganda having latrines [17]. Our findings support the lower estimates, with a large number of people having to perform open defecation due to lack of facilities, mirroring other Uganda research [64] and research in Tanzania and Nigeria [67,68]. This leads to the continued practice of activities that put both themselves and others at risk of catching schistosomiasis [9]. Continued open defecation and inadequate containment of faeces remain strong barriers to transmission prevention and disease control.

## Recommendations to help the people in the studied communities prevent schistosomiasis

Based on our findings, we recommend improved and more frequent health education and sensitization at a micro level so that accurate information can reach every individual to improve awareness of schistosomiasis using social and behavioural change communication [69]. This will enable the further dissemination of contextualised messaging and health education [45, 64] tailored towards specific community needs such as discussing the differences of the symptoms between diseases including elephantiasis, schistosomiasis and any other locally relevant diseases, and eliminating confusion surrounding associations between schistosomiasis and *ekidada*. Such improved messaging could occur through schools [55], consistent accurate radio

announcements [55], and/or accompanied by showing videos [69] portraying the life cycle of schistosomiasis and parasites in the water and how they affect people.

In addition, we recommend health education that involves reaching out to individuals through communal gatherings [64]. A further advantage of gathering people is that they can ask questions about what they seem not to understand. Since gathering people in one place is not easy this can be complemented by public radio announcements to reach people who did not attend. Public radio seems to be more effective than mainstream radio as it can reach out to everyone within a community including those with no personal radios [70,71].

There is a need to refine and improve messaging for all endemic NTDs [55]. We need to emphasise all visible symptoms as well as more subtle personal ones and encourage a pragmatic approach whereby people who seek treatment for *ekidada*, or other problems, also take praziquantel. This may be best achieved by working with traditional healers to encourage this pragmatic approach as demonstrated in Botswana during the emergence of HIV where traditional healers would refer their patients to health centres if the symptoms they saw in participants were likely to be linked with HIV [72].

We recommend improving availability, and accessibility of WASH facilities including safe water, to reduce everyday contact with lake water [64,68]. Careful consideration needs to be given to the siting of affordable, acceptable, reliable alternative water sources like placing piers at the lake to reduce submergence for those who prefer, or need to continue, collecting water from the lake, or providing a free water filtration system to reduce parasite load [73].

Construction and maintenance of permanent public latrines are needed in the communities to help curb the practice of open defecation [64] and improve the containment of faeces when latrines are used [74]. However, providing latrines alone may not be enough because lack of access is not the only barrier to uptake. Some participants reported that it is a taboo for women of some tribes to defecate in a latrine, and many fishermen described the lake as too large to be contaminated by faeces and considered open defecation as part of their habits and methods to improve fishing success. Therefore, even when latrines are provided, locally informed education and behavioural nudges may be required.

Finally, we recommend the government introduce development programs to empower people economically [75] because a crucial driver of reported water contact was that people could not afford safe water and other preventive methods. People widely acknowledge that preventive methods cannot be put in place by themselves without government support. Therefore, a combined approach of the district governments working with communities to improve sanitation infrastructure may be the most successful intervention, raising awareness at the same time as improving infrastructure.

## Conclusion

In conclusion, our findings indicate a range of understandings and misunderstandings around schistosomiasis. The perceptions presented by participants about schistosomiasis and its transmission profoundly shaped what people did to reduce their risk, which greatly affected control practices such as avoiding contact with infected water, latrine use, and praziquantel uptake. It is evident that these complex combined perceptions contribute to the persistence of schistosomiasis in the community, influencing infection and re-infection and hindering the WHO targets of elimination as a public health problem by 2030.

## Supporting information

**S1 File. Raw data extracted and anonymised data after coding.**
(DOCX)

## Acknowledgments

The authors would like to thank the District Officials of Mayuge, Juma Nabonge the District NTD focal person, study participants, VHTs, beach management committees and community leaders of Bugoto, Bwondha and Musubi for their time and help to facilitate and take part in this research. We thank Hassan Ssenyonga for his commitment to driving us to the field, Dr Suzan Trienekens for making the maps and Prof. Sally Wyke who helped to co-design the study.

## Author Contributions

**Conceptualization:** Lucy Pickering, Janet Seeley, Poppy H. L. Lamberton.

**Data curation:** Lazaaro Mujumbusi, Edith Nalwadda, Agnes Ssali, Lucy Pickering, Poppy H. L. Lamberton.

**Formal analysis:** Lazaaro Mujumbusi, Edith Nalwadda, Agnes Ssali, Lucy Pickering, Keila Meginnis.

**Funding acquisition:** Lucy Pickering, Janet Seeley, Poppy H. L. Lamberton.

**Investigation:** Lazaaro Mujumbusi, Edith Nalwadda.

**Methodology:** Lazaaro Mujumbusi, Edith Nalwadda, Agnes Ssali, Lucy Pickering, Janet Seeley, Poppy H. L. Lamberton.

**Project administration:** Lazaaro Mujumbusi, Edith Nalwadda, Agnes Ssali, Lucy Pickering, Janet Seeley, Keila Meginnis, Poppy H. L. Lamberton.

**Resources:** Janet Seeley, Poppy H. L. Lamberton.

**Supervision:** Agnes Ssali, Lucy Pickering, Janet Seeley, Keila Meginnis, Poppy H. L. Lamberton.

**Validation:** Lazaaro Mujumbusi, Edith Nalwadda, Agnes Ssali, Lucy Pickering, Janet Seeley, Keila Meginnis, Poppy H. L. Lamberton.

**Visualization:** Poppy H. L. Lamberton.

**Writing – original draft:** Lazaaro Mujumbusi.

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
