## [Decision Letter · Decision Letter 0]

23 Sep 2022

Dear Mr Mujumbusi,

Thank you very much for submitting your manuscript "Understanding perception of schistosomiasis and its control  among lakeshore communities in Mayuge, Uganda." for consideration at PLOS Neglected Tropical Diseases. As with all papers reviewed by the journal, your manuscript was reviewed by members of the editorial board and by several independent reviewers. The reviewers appreciated the attention to an important topic. Based on the reviews, we are likely to accept this manuscript for publication, providing that you modify the manuscript according to the review recommendations. 

Sincerely,

Uwem Friday Ekpo, PhD

Academic Editor

Francesca Tamarozzi

Section Editor

Reviewer's Responses to Questions

**Key Review Criteria Required for Acceptance?**

**Methods**

-Are the objectives of the study clearly articulated with a clear testable hypothesis stated?

-Is the study design appropriate to address the stated objectives?

-Is the population clearly described and appropriate for the hypothesis being tested?

-Is the sample size sufficient to ensure adequate power to address the hypothesis being tested?

-Were correct statistical analysis used to support conclusions?

-Are there concerns about ethical or regulatory requirements being met?

Reviewer #1: (No Response)

Reviewer #2: (No Response)

Reviewer #3: Clearly articulated methodology

**Results**

-Does the analysis presented match the analysis plan?

-Are the results clearly and completely presented?

-Are the figures (Tables, Images) of sufficient quality for clarity?

Reviewer #1: (No Response)

Reviewer #2: (No Response)

Reviewer #3: Results clear

**Conclusions**

-Are the conclusions supported by the data presented?

-Are the limitations of analysis clearly described?

-Do the authors discuss how these data can be helpful to advance our understanding of the topic under study?

-Is public health relevance addressed?

Reviewer #1: (No Response)

Reviewer #2: (No Response)

Reviewer #3: Conclusions are supported by the data presented and of public health relevance

**Editorial and Data Presentation Modifications?**

Reviewer #1: (No Response)

Reviewer #2: (No Response)

Reviewer #3: Authors should consider making the following changes:

Lines 77-79: to move towards the end of Introduction section, can consider merging it with Lines 162-163

Lines 80-85: references missing

Lines 81-82: it should read "actively penetrate the skin and became schistosomule. These enter the circulatory system and migrate to the liver where they develop into adult worms, pair off, and sexually reproduce." 

Line 103: consider putting recent data, as these one are 5 years and older.

Lines 198 and 203-204: the acronym "WaSH" look different, please correct.

Line 276: remove "s" after the word "utensils".

Lines 289-290: consider moving this statement to Discussion

Lines 351-352: consider moving this statement to Discussion

Line 382: correct the spelling for Praziquantel.

Lines 572-576 and 593-594: consider moving these statements to Discussion.

Lines 689-740: consider including in the recommendation "improving availability and accessibility of WaSH facilities like safe water, etc", as this is in the results nd discussion, not coming out clearly in the recommendations.

**Summary and General Comments**

Reviewer #1: This is a very interesting area to explore. As the authors say, Uganda has received many rounds of treatment over the last 15 years with promising progress in some areas, but also some persistent areas of infection, and even increases, including in Mayuge. This type of qualitative, social-science approach, such as using the rapid ethnographic appraisal, is very welcome. Treatment coverage is worrying low here.

“35.3% of the population eligible to take praziquantel reported never to have taken it, despite over 10 years of MDA in this community” is an incredible statistic.

I found this article to be very well written, clear, comprehensive, and very interesting. I only have some minor suggestions / comments.

Comments

• As well as their knowledge about infection and symptoms, did the communities see SCH as a problem?

• Did you record how many of the interviewees had taken PZQ? At last round or ever?

• Out of interest, what was the intensity of infection in these areas? Presumably high as well. Were there a lot of people harboring heavy infections?

Minor / Editorial

• Line 83 – haematobium should be in italics

• Line 110 – remove extra comma

• Line 276 – check typo between ‘utensils’ and ‘other household activities’

• Figure 4 – typo in title

Reviewer #2: (No Response)

Reviewer #3: This is study is very relevant and provide important findings on the perceptions of Schistosoma mansoni disease by people living in endemic areas along Lake Victoria.

The findings and recommendations are very vital for the health officers in the national control program as they implement strategies to control schistosomiasis and move towards eliminating it as a public health problem by 2030.

PLOS authors have the option to publish the peer review history of their article (what does this mean?). If published, this will include your full peer review and any attached files.

Reviewer #1: Yes: Michael French

Reviewer #2: No

Reviewer #3: No

Figure Files:

Data Requirements:

Reproducibility:

References

---

## [Editor Report · Decision Letter 1]

7 Dec 2022

Dear Mr Mujumbusi,

Thank you very much for submitting your manuscript "Understanding perceptions of schistosomiasis and its control  among highly endemic lakeshore communities in Mayuge, Uganda." for consideration at PLOS Neglected Tropical Diseases. As with all papers reviewed by the journal, your manuscript was reviewed by members of the editorial board and by several independent reviewers. The reviewers appreciated the attention to an important topic. Based on the reviews, we are likely to accept this manuscript for publication, providing that you modify the manuscript according to the review recommendations. 

Dear Authors,

The correct acronym for Water, Sanitation and Hygiene is WASH not WaSH as used throughout the manuscript. WASH is a UNICEF approved acronym. Please revised the acronym throughout your manuscript including the reference section. Obviously, WaSH is not the same as WASH.

Sincerely,

Uwem Friday Ekpo, PhD

Academic Editor

Francesca Tamarozzi

Section Editor

Dear Authors,

The correct acronym for Water, Sanitation and Hygiene is WASH not WaSH as used throughout the manuscript. WASH is a UNICEF approved acronym. Please revised the acronym throughout your manuscript including the reference section. Obviously, WaSH is not the same as WASH.

Figure Files:

Data Requirements:

Reproducibility:

References

---

## [Editor Report · Decision Letter 2]

15 Dec 2022

Dear Mr Mujumbusi,

We are pleased to inform you that your manuscript 'Understanding perceptions of schistosomiasis and its control  among highly endemic lakeshore communities in Mayuge, Uganda.' has been provisionally accepted for publication in PLOS Neglected Tropical Diseases.

Best regards,

Uwem Friday Ekpo, PhD

Academic Editor

Francesca Tamarozzi

Section Editor

---

## [Editor Report · Acceptance letter]

16 Jan 2023

Dear Mr Mujumbusi,

We are delighted to inform you that your manuscript, "Understanding perceptions of schistosomiasis and its control  among highly endemic lakeshore communities in Mayuge, Uganda.," has been formally accepted for publication in PLOS Neglected Tropical Diseases.

Best regards,

Shaden Kamhawi

co-Editor-in-Chief

Paul Brindley

co-Editor-in-Chief
